# Vertebra partitioning with thin-plate spline surfaces steered by a convolutional neural network

**Nikolas Lessmann**                                        NIKOLAS.LESSMANN@RADBOUDUMC.NL
*Radboud University Medical Center, Nijmegen, The Netherlands*

**Jelmer M. Wolterink** and **Majd Zreik** and **Max A. Viergever**
*University Medical Center Utrecht, The Netherlands*

**Bram van Ginneken**
*Radboud University Medical Center, Nijmegen, The Netherlands*

**Ivana Išgum**
*University Medical Center Utrecht, The Netherlands*

## Abstract

Thin-plate splines can be used for interpolation of image values, but can also be used to represent a smooth surface, such as the boundary between two structures. We present a method for partitioning vertebra segmentation masks into two substructures, the vertebral body and the posterior elements, using a convolutional neural network that predicts the boundary between the two structures. This boundary is modeled as a thin-plate spline surface defined by a set of control points predicted by the network. The neural network is trained using the reconstruction error of a convolutional autoencoder to enable the use of unpaired data.

**Keywords:** thin-plate splines, shape analysis, vertebra partitioning, autoencoder

## 1. Introduction

Splines are commonly used for interpolation and, for that purpose, have recently also been integrated into convolutional neural networks (CNNs), for example in spatial transformer networks (Jaderberg et al., 2015) or in image registration frameworks (de Vos et al., 2019). However, splines such as thin-plate splines (TPS) can not only be used to interpolate image values, but can also be used to represent a smooth surface, for instance the boundary between two structures. We leverage this property in a method for partitioning of segmentation masks of vertebrae into two substructures, namely the vertebral body and the vertebral posterior elements. The approach is based on a CNN that predicts the location and shape of a TPS surface by predicting the location of a set of control points that define the surface. The CNN is trained with the help of a convolutional autoencoder, which serves as a shape model of the vertebral body, to enable training the CNN with unpaired data (Figure 1). We trained the method with vertebra segmentations from chest CT images and vertebral body segmentations from lumbar spine MR images, and evaluated the partitioning results on a set of chest CT scans with both vertebra and vertebral body reference segmentations.

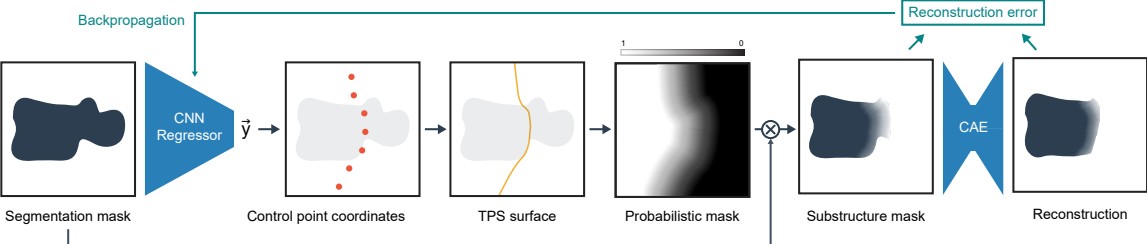

Figure 1: Flowchart illustrating the training process. Vertebra segmentation masks are fed into a regression CNN that predicts the y-coordinates of several control points. These define a TPS surface, from which a probabilistic mask is derived, which is applied to the vertebra masks to obtain soft vertebral body masks. These are passed through a convolutional autoencoder (CAE), trained to represent a shape model of the substructure. The reconstruction error is used to update the CNN.

## 2. Methods

Training a CNN that predicts the location and shape of a TPS surface by predicting the coordinates of its control points requires calculating the spline coefficients as part of the forward pass. This involves solving a system of linear equations, which is typically not a differentiable operation. However, by choosing a fixed grid of control points in a plane and letting the network predict only the third coordinate, i.e., the height of the surface relative to that plane, the spline coefficients can be calculated in a differentiable manner. Specifically, the 2D grid coordinates of the control points as well as terms depending on the in-plane distance of the control points can be accumulated in a single matrix of which a pseudo-inverse can be precomputed before training the network (Bookstein, 1989). During forward passes through the network, the spline coefficients can then be calculated with a single matrix-vector multiplication.

The predicted continous TPS surface is used to mask out the vertebral posterior elements, but could likewise be used to mask out the vertebral body. We consider the distance of each voxel of the input vertebra segmentation mask to the TPS surface, and use the sigmoid function to convert these distances into a probabilistic mask with values in $[0, 1]$. This mask is multiplied with the vertebra segmentation mask, resulting in a soft segmentation of the vertebral body (Figure 1). A loss function for training the network could be defined with these soft masks and corresponding reference segmentations of the vertebral bodies. However, this requires a set of paired segmentation masks, where both the entire vertebrae as well as the vertebral bodies have been segmented in the same scans. We instead address the case were segmentation masks of vertebrae and vertebral bodies are available, but from different scans (and possibly different subjects or modalities). This enables the combination of multiple datasets, for example from segmentation challenges.

We train a convolutional autoencoder (CAE) with segmentation masks of vertebral bodies. This CAE therefore learns a shape model of vertebral bodies (Oktay et al., 2018).

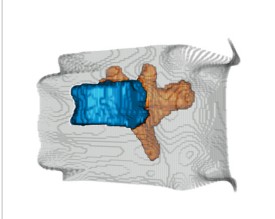 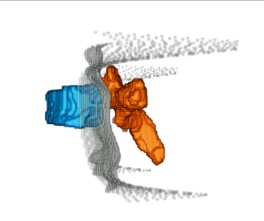 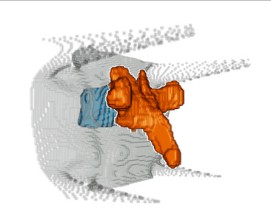 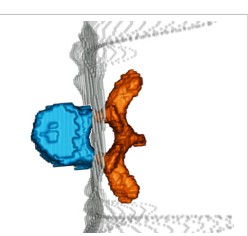

Figure 2: Partitioning result as three-dimensional renderings. The vertebral body partition is displayed in blue, the posterior partition in orange and the TPS surface in gray. Note that the predicted surfaces are not smooth in the parts where there is never any vertebral bone and hence no gradient information, i.e., at the very top and bottom of the volume.

The CAE is trained beforehand and is fixed while training the TPS-CNN. Assuming that the CAE will reconstruct segmentation masks well only if they are very similar to vertebral body segmentation masks, and that TPS surfaces leading to an incorrect partitioning will therefore produce vertebral body masks that the CAE cannot reconstruct well, we use the reconstruction error of the CAE to update the TPS-CNN (Figure 1).

## 3. Results and Discussion

We used a set of 6247 vertebra segmentation masks obtained from 580 low-dose chest CT scans and 1193 vertebral body segmentation masks obtained from 314 lumbar spine MR scans. There was no overlap in patients between the two datasets. The segmentation masks were automatically obtained with an iterative segmentation method (Lessmann et al., 2019) and were manually reviewed for segmentation errors. Input volumes to both the TPS-CNN and the CAE had a size of $128 \times 128 \times 128$ voxels at 1 mm isotropic resolution. In six chest CT scans, the vertebral bodies were manually segmented and the partitioning results were evaluated with five of these annotations (50 vertebrae), the sixth was used for validation during training. We computed the Dice coefficient as well as the Hausdorff distance for networks trained with different grid spacing, i.e., different number of control points.

For 64, 100, 256 and 1024 control points, the Dice coefficient ranged from $98.9 \pm 1.1 \%$ to $99.3 \pm 0.5 \%$. The Hausdorff distance ranged from $4.1 \pm 2.3$ mm to $3.5 \pm 1.8$ mm. Visual inspection of the results (Figure 2) revealed that the TPS surface was in most cases placed in approximately the right location, which, however, is often also difficult to determine manually in low-dose chest CT scans. The predicted surfaces were well adapted to various spinal curvatures and to a range of vertebrae (T2–T11), differing in size and shape.

In conclusion, while a more thorough evaluation is still needed, these initial results demonstrate that a CNN can be used for a partitioning task not only by labeling voxels, but also by steering a smooth and continuous surface representing the boundary between the partitions. We additionally demonstrated that a CAE can be used to train such an approach by learning a model of plausible shapes and using the reconstruction error to express the plausibility of partitioning results.

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
