# OpenReview forum: "Vertebra partitioning with thin-plate spline surfaces steered by a convolutional neural network"
_MIDL.io/2019/Conference/Abstract — MIDL Abstract 2019_

### Official Review · AnonReviewer2 · 2019-04-25
**Interesting loss, will make nice application poster**

**Rating:** 4
**Confidence:** 3

**Review:**

This is a well written abstract, describing a CNN regressor that places a bunch of points in space to define a surface that partitions a segmentation into 2 subregions. I find most interesting the use of the reconstruction error of a (fixed) convolutional autoencoder on one of the two subregions as loss function for the CNN. I think this poster could lead to some good discussion, and I am looking forward to seeing more extensive experiments (e.g., comparison with supervised approaches) in the full version.

---

### Official Review · AnonReviewer1 · 2019-04-29
**Thin plate spline surfaces for detecting boundary between vertebra structures**

**Rating:** 3
**Confidence:** 2

**Review:**

The thin-plate spline is used in this paper to partition segmentation mask from CNN into two vertebrae structures.

The paper proposes a novel method to estimate the parameters of TPS surface by predicting the height of the surface relative to a fixed plane. A CNN regressor produces these spline coefficients. The experiments results are promising. A CAE at the end is used to show the effect of partitioning with the reconstruction error.

The paper is well written.

---

### Decision · Program_Chairs · 2019-05-06
**Acceptance Decision**

Accept